# Eating breakfast and avoiding late-evening snacking sustains lipid oxidation

**Kevin Parsons Kelly**[1], **Owen P. McGuinness**[2], **Maciej Buchowski**[3], **Jacob J. Hughey**[4], **Heidi Chen**[5], **James Powers**[6,7], **Terry Page**[1], **Carl Hirschie Johnson**[1,2]*

**1** Department of Biological Sciences, Vanderbilt University, Nashville, Tennessee, United States of America, **2** Department of Molecular Physiology and Biophysics, Vanderbilt University School of Medicine, Nashville, Tennessee, United States of America, **3** Division of Gastroenterology, Hepatology, & Nutrition, Vanderbilt University Medical Center, Nashville, Tennessee, United States of America, **4** Department of Biomedical Informatics, Vanderbilt University Medical Center, Nashville, Tennessee, United States of America, **5** Department of Biostatistics, Vanderbilt University Medical Center, Nashville, Tennessee, United States of America, **6** Department of Medicine, Vanderbilt University Medical Center, Nashville, Tennessee, United States of America, **7** Tennessee Valley Healthcare System Geriatric Research, Education, and Clinical Center, Nashville, Tennessee, United States of America

* carl.h.johnson@vanderbilt.edu

**Data Availability Statement:** All relevant data are within the paper and its Supporting Information files.

## Abstract

Circadian (daily) regulation of metabolic pathways implies that food may be metabolized differentially over the daily cycle. To test that hypothesis, we monitored the metabolism of older subjects in a whole-room respiratory chamber over two separate 56-h sessions in a random crossover design. In one session, one of the 3 daily meals was presented as breakfast, whereas in the other session, a nutritionally equivalent meal was presented as a late-evening snack. The duration of the overnight fast was the same for both sessions. Whereas the two sessions did not differ in overall energy expenditure, the respiratory exchange ratio (RER) was different during sleep between the two sessions. Unexpectedly, this difference in RER due to daily meal timing was not due to daily differences in physical activity, sleep disruption, or core body temperature (CBT). Rather, we found that the daily timing of nutrient availability coupled with daily/circadian control of metabolism drives a switch in substrate preference such that the late-evening Snack Session resulted in significantly lower lipid oxidation (LO) compared to the Breakfast Session. Therefore, the timing of meals during the day/night cycle affects how ingested food is oxidized or stored in humans, with important implications for optimal eating habits.

## Introduction

Developed countries are experiencing an epidemic of obesity that leads to many serious health problems, foremost among which are increasing rates of type 2 diabetes, metabolic syndrome, cardiovascular disease, and cancer. While weight gain and obesity are primarily determined by diet and exercise, there is tremendous interest in the possibility that the daily timing of eating might have a significant impact upon weight management [1–3]. Many physiological processes display day/night rhythms, including feeding behavior, lipid and carbohydrate metabolism, body temperature, and sleep. These daily oscillations are controlled by the circadian clock, which is composed of an autoregulatory biochemical mechanism that is expressed in tissues

**Funding:** This study was supported by a Vanderbilt Discovery Grant (to TP), the Vanderbilt Institute for Clinical and Translational Research (VICTR) award ID# VR9806 (to MB), the Vanderbilt Diabetes Research and Training Center (through the Metabolic Physiology Shared Resource supported by P60-DK020593; to OPM), the National Institute of General Medical Sciences (R35 GM124685 to JJH), and the National Institute of Neurological Disorders and Stroke (R01 NS104497 to CHJ). The funders had no role in study design, data collection and analysis, decision to publish, or preparation of the manuscript.

**Competing interests:** The authors have declared that no competing interests exist.

**Abbreviations:** BMI, Body Mass Index; CBT, core body temperature; CO, carbohydrate oxidation; LO, lipid oxidation; MR, metabolic rate; RER, respiratory exchange ratio; SCN, suprachiasmatic nuclei; TEF, thermic effect of food.

throughout the body and is coordinated by a master pacemaker located in the suprachiasmatic nuclei of the brain (aka the SCN [1,4]). The circadian system globally controls gene expression patterns so that metabolic pathways are differentially regulated over the day, including switching between carbohydrate and lipid catabolism [1,3,5–9]. Therefore, ingestion of the same food at different times of day could lead to differential metabolic outcomes, e.g., lipid oxidation (LO) versus accumulation; however, whether this is true or not is unclear.

Nonoptimal phasing of the endogenous circadian system with the environmental day/night cycle has adverse health consequences. Shiftworkers are a particularly cogent example because their work schedule disrupts the optimal relationship between the internal biological clock and the environmental daily cycle, and this disruption leads to well-documented health decrements [6,9–13]. A contributing culprit that is often implicated in shiftwork's temporal disruption is the disturbance of eating patterns and preferences. In nonhuman mammals, a persuasive literature demonstrates that manipulating the timing of feeding relative to biological clock phase effectively controls obesity [1,14,15]. In particular, mice fed a high-fat diet on a restricted schedule maintain a healthy weight when fed only during their active phase but become obese if the high-fat diet is present during the inactive phase, even though the long-term caloric intake and locomotor activity levels are comparable between day- versus night-fed mice [14,15].

Can the timing of eating relative to our circadian cycle of metabolism and sleep also help to regulate lipid metabolism and body weight in humans? Eating late in the day is correlated with weight gain [16], and there is an oft-discussed debate whether skipping breakfast versus dinner reaps weight-control benefits [3,17, 18,19]. While many factors can influence the timing of eating in everyday life [2,3,20,21], we decided to take an experimental approach to test the metabolic consequences of a straightforward exchange of equivalent nutritional intake between early morning (8 AM) and bedtime (10 PM). While it is not feasible to do the 12-h reversal of feeding time that was tested with mice [14] because it would disrupt the consolidated sleep episode of humans, we focused upon a 4.5 h shift of feeding in which human subjects ate either an approximately 700 kcal breakfast or an equivalent approximately 700 kcal late-evening snack meal. Not only are these two feeding schedules experimentally tractable for a human study, they are also commonly practiced by humans in everyday life (i.e., "skipping breakfast" and/or "late-evening snacking"). For each feeding schedule, we monitored the metabolism of our subjects for a 56-h stint in a whole-room calorimetry chamber to continuously measure their metabolic rate (MR), respiratory exchange ratio (RER), carbohydrate oxidation (CO), and LO.

Previous studies of human metabolism for shorter monitoring periods (approximately 24 h) suggested that overall 24-h energy expenditure was not significantly affected by either breakfast skipping or a late dinner [22,23]); however, those prior studies were performed on healthy Asian young adults of optimal Body Mass Index (BMI) (18.5–25 kg/m$^2$) for only approximately 24 h, which is inadequate to study a phenomenon based on circadian rhythmicity. Moreover, while differences in blood glucose levels were reported in those studies between breakfast-skipping or late-dinner sessions, LO was either not affected (breakfast skipping [23]) or counterintuitively enhanced (late dinner [22]). In this investigation, we monitored older Caucasian adults (aged 50 or above) of varying BMI because we reasoned they are more representative of populations at risk for metabolic disorders in many developed countries than are young and healthy adults. Each subject underwent two 56-h (2.5-d) sessions in a whole-room respiratory chamber, and with a randomized crossover design, we compared the energy expenditure (MR) and RER of each subject when given a scheduled breakfast, lunch, and dinner (Breakfast Session) versus when they were given a lunch, dinner, and a late-evening snack (Snack Session).

While overall 24-h energy expenditure was similar in this group of older subjects, RER was significantly different between the two sessions. We anticipated that daily differences in physical activity, sleep disruption, or core body temperature (CBT) might lead to differential metabolism as reflected in the RER. Unexpectedly, however, our data demonstrated that even though the total daily energy and nutrient intake was equivalent between the sessions, switching the daily timing of a nutritionally equivalent 700-kcal meal from a "breakfast" to a "late-evening snack" had a significant impact upon carbohydrate and lipid metabolism such that nocturnal CO was favored at the expense of LO when subjects ate the 700-kcal meal as a late-evening snack. Therefore, the daily cycle of metabolism and nutrient availability switches substrate preference so that the cumulative net LO is altered by the timing of meals.

## Results

We studied the metabolism of human subjects by indirect calorimetry under continuous monitoring in Vanderbilt University's Human Metabolic Chamber. During each visit, the minute-by minute oxygen consumption ($VO_2$), carbon dioxide production ($VCO_2$), actigraphy, and CBT of the subjects were continuously measured, with subsequent calculation of RER ($VCO_2/VO_2$), MR, CO, and LO. The subjects slept and ate in the metabolic chamber and were allowed only two brief (20-min) episodes per day outside the chamber: once about 10:00 AM to take a quick shower and once about 3:00 PM to take a brief nonstrenuous walk. Each subject was monitored for two full-duration 56-h experiments that compared differences in the timing of their meals. We use the terms "breakfast" to mean a meal at 08:00–09:00, "lunch" at 12:30–13:30, "dinner" at 17:45–18:15, and a late-evening "snack" meal at 22:00–23:00. Consequently, in the "Breakfast Session" (Fig 1A), the subjects had breakfast, lunch, and dinner, with an approximately 13.75-h fast from 6:15 PM (end of dinner) to 8:00 AM (breakfast). In the "Snack Session," subjects only had a cup of tea or coffee (without sugar or creamer) at breakfast time, and their first meal was lunch (Fig 1A). Then, the "snack meal" was served at 10:00 PM just before sleep (lights off), and the subjects fasted approximately 14 h until lunch (10:30 PM–12:30 PM the next day). The breakfasts and the snacks had equivalent nutritional and caloric values of approximately 700 kcal; the breakfast on day 2 was identical to the snack on day 2, and the breakfast on day 3 was identical to the snack on day 3 (S1A Table). Therefore, the meals served to subjects during the Breakfast Session had equivalent energy and nutrient content as in the Snack Session over the 24-h day (see S1 Table for detailed nutritional information). All subjects completed both sessions in this crossover experiment, which allowed pairwise comparison of their data. Our study is distinguished from the earlier metabolic chamber studies of meal timing in humans [22,23] by the crossover design of our protocol and the fact that we studied older subjects of various BMIs (51–63 years old, BMI 22.2–33.4; see Table 1), who may be less resilient to metabolic perturbations for energy expenditure than are younger subjects with BMIs of 20–25 [24,25]. Moreover, our Breakfast versus Snack Sessions had essentially the same duration of daily fasting (13.75–14 h) to avoid the confounding factor of differential fasting durations found in other studies [21].

As illustrated in Fig 1B, subjects in the Breakfast Session that included breakfast and a fast throughout the time interval from dinner to the following breakfast (6:30 PM to 8:00 AM) exhibited a strong daily rhythm of RER (aka respiratory quotient, calculated as $VCO_2/VO_2$ [26,27]). RER values close to 0.7 indicate LO, while values of approximately 1.0 indicate almost exclusive CO. The average RER of this diet is similar to that of typical diets in the United States of America (approximately 0.85); it includes a mixture of lipids, protein, and carbohydrates [26,27]. The RER of subjects in the Breakfast Session was low throughout the lights-off interval (indicating primarily lipid catabolism during sleep) and high during the active daytime

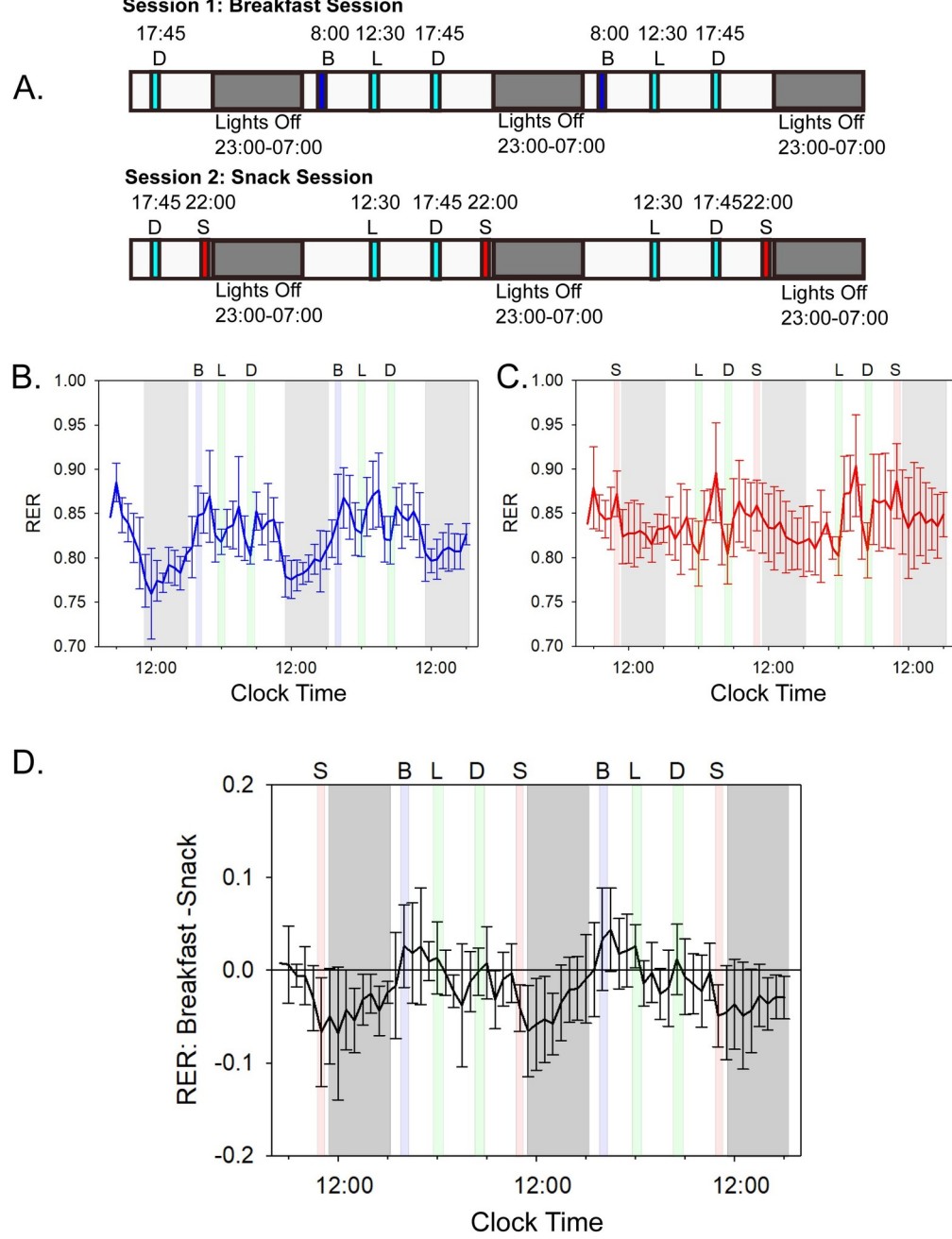

**Fig 1. Chamber schedule and the impact of meal timing on subjects' RERs.** See S3 Fig and S2A Table and S1 Data for underlying data and analyses. (A) Protocol for "Breakfast" Session versus "Snack" Session. Subjects experienced two separate 56-h continuous sessions with constant metabolic monitoring by indirect calorimetry, each session lasting 56 h. The Breakfast Session included a breakfast (B), lunch (L), and dinner (D), while the Snack Session contained a lunch, dinner, and late-evening snack meal (S). The late-evening snacks were of equivalent caloric and nutritional value to the breakfast meals (approximately 700 kcal; see S1 Table for details). Note from S2 Fig that the daily phasing of sleep for the subjects prior to entry into the metabolic chamber was the same as the "lights-off" interval during the 56-h time course, so the subjects did not experience a phase shift of their daily cycle when they entered the experimental conditions. (B) Breakfast Session: blue line indicates the average hourly RER over the entire 56-h time course among all 6 subjects when a breakfast, lunch, and dinner were presented. Error bars are the standard deviation. Letters indicate time and type of meals, and gray shaded areas indicate the lights-off periods. Green shaded areas indicate meals that were given at the same time in both Breakfast and Snack Sessions (lunch and dinner). Blue shaded areas indicate when breakfast was given, and gray shading indicates the lights-off period. See S3 Fig for data of all subjects individually. (C) Snack Session: the red line indicates the average hourly RER over the entire 56-h time course

among all subjects when a lunch, dinner, and late-evening snack were presented. Red shaded areas indicate when late-evening snacks were given. Breakfasts and late-evening snacks contained the same number of calories and the same lipid, carbohydrate, and protein content (S1A and S1B Table). Error bars are the standard deviation ($n$ = 6). (D) Average difference in RER over the entire 56-h time course for the Breakfast Session subtracted from the Snack Session. Deviation from zero (horizontal black line) indicates where differences in RER occurred between subjects. Error bars indicate standard deviation in the differences. In Panels B, C, and D, times of meals are indicated by letters (B = breakfast, L = lunch, D = dinner, S = late-evening snack), and gray areas are lights-off (sleep) intervals. Mealtimes are shaded as in panels A–C; breakfasts and snacks occurred only in their respective sessions. All RER data were collected minute by minute, and in this figure, the minute-by-minute data were binned and averaged for all 60 values within an hour. Abscissa are clock time. RER, respiratory exchange ratio.

(indicating primarily carbohydrate and protein catabolism). Therefore, humans share with other mammals a daily rhythm of substrate metabolism as assessed by indirect calorimetry [28,29]. However, in the Snack Session, the metabolism of the same subjects was not as strongly rhythmic and displayed a lower amplitude rhythm of RER that did not drop into a largely lipid-catabolic mode (Fig 1C). When the difference between the RER on the Breakfast versus the Snack Sessions is calculated as a function of daily time, the most significant difference was noted during the inactive sleep phase, during which LO predominates in the Breakfast Session, while the RER remains high in the Snack Session (Fig 1D).

We initially predicted that the session-dependent RER patterns were due to differences between the sessions in physical activity, sleep disruption, CBT, or the phasing/amplitude of the circadian clock. However, none of these parameters were different between the sessions. Actigraphy confirmed that the subjects' overall activity levels did not differ significantly between the two sessions (S1A and S1B Fig, $p$ = 0.538). Moreover, actigraphy can provide an assessment of restlessness during sleep [30] and by this criterion, the sleep quality was equivalent between the Breakfast versus Snack Sessions (S1A and S1B Fig). The daily rhythm of the CBT, which is frequently used as a marker of the central circadian clock in humans [9,31], did not show significant differences in the phasing or amplitude of their CBT rhythms between the two sessions (S1C and S1D Fig, $p$ = 0.218). Moreover, the circadian rhythm of overall MR [8] was not different in phase or amplitude between the two sessions (Fig 2B, $p$ = 0.11). Therefore, neither differences in CBT (S1 Fig) nor the thermic effect of food (TEF; see below) were responsible for the session-dependent RER differences. The phasing or amplitude of the daily rhythms of master clock markers (plasma melatonin and cortisol) is also not responsible, as reported by Wehrens and colleagues, who found no differences in those rhythms in a meal timing study using a similar protocol to ours [32]. Finally, our subjects kept a regularly timed sleep/wake cycle prior to the metabolic chamber experiments so that their internal rhythm was in phase with the light/dark cycle during the 56-h experiment (S2 Fig and Table 1). These results suggest that the change in meal timing altered the RER rhythm (i.e., its amplitude) without changing overall activity, sleep quality, body temperature, or the phase relationship between circadian rhythms and the daily light/dark schedule.

**Table 1. Subjects involved in this study.**

| ID | Sex | Age (Years) | Weight (kg) | Height (cm) | BMI | Self-Reported Bedtime | Self-Reported Wake Time | First Session Meal Plan |
|----|-----|-------------|-------------|-------------|------|----------------------|------------------------|------------------------|
| 1 | M | 61 | 106.8 | 187.96 | 30.2 | 23:00 | 7:00 | Breakfast |
| 2 | M | 58 | 101.36 | 177.8 | 32.1 | 22:30 | 6:00 | Snack |
| 3 | M | 51 | 73.9 | 184 | 21.8 | 23:15 | 5:30 | Snack |
| 4 | F | 57 | 68.2 | 173 | 22.8 | 22:30 | 6:30 | Breakfast |
| 5 | F | 63 | 62.7 | 168 | 22.2 | 22:00 | 7:00 | Breakfast |
| 6 | M | 54 | 85 | 180 | 26.2 | 22:30 | 6:15 | Snack |

**Abbreviations:** BMI, Body Mass Index.

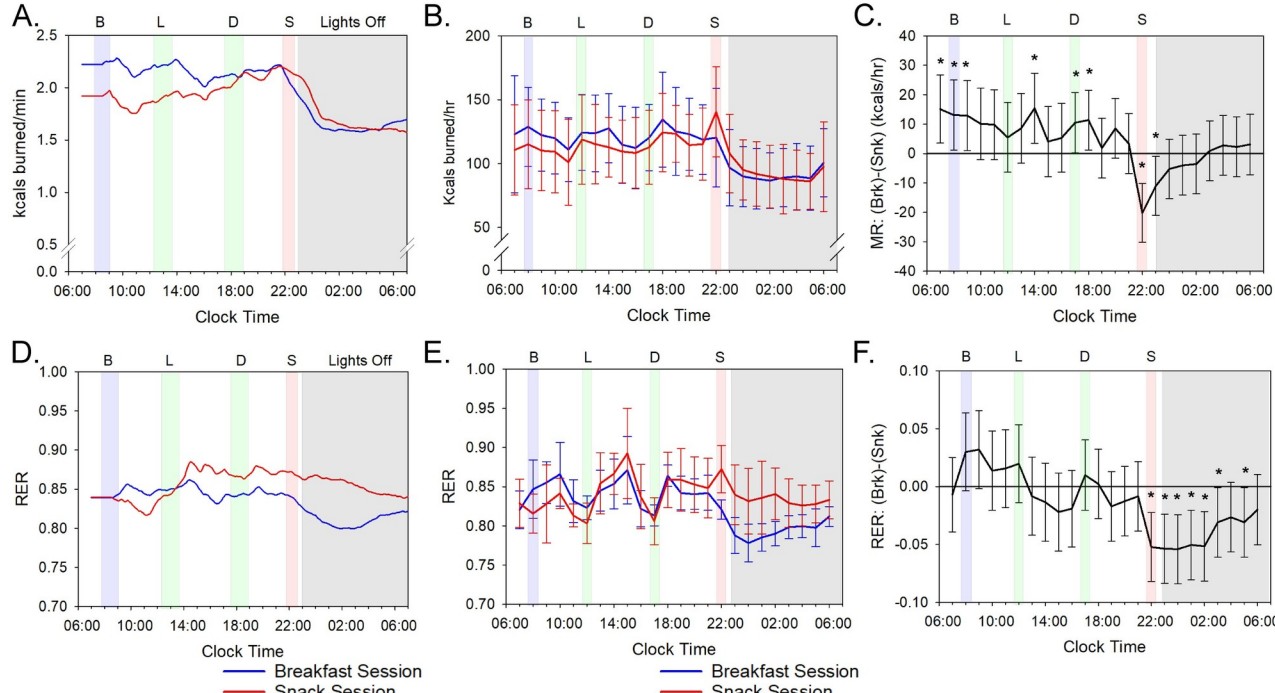

**Fig 2. MRs and RER values.** See S3 and S4 Figs and S2A and S2D Table for underlying data and analyses. (A) MR by indirect calorimetry for a representative participant (Subject #3). The data for Subject #3 are plotted as a moving average using 180 data points (= 3 h) after aligning all time points to clock time and integrated on a 24-h scale. (B) Average MR for all subjects plotted modulo-24 h. Data were averaged into 1-h bins with error bars indicating standard deviation. See S4 Fig for data of all subjects' MRs individually plotted. (C) Average hourly pairwise comparison of (breakfast − snack) MR values for all subjects. Error bars indicate 95% confidence intervals and values are based on a mixed-model analysis. Asterisks indicate significant differences (*p*-value < 0.05) between breakfast and snack values for the indicated 1-h bins. See S2D Table for the hour-by-hour statistical comparison of the Breakfast versus the Snack Sessions. (D) RER ($VCO_2/VO_2$) by indirect calorimetry of a representative individual (subject #3). The data for Subject #3 are plotted as a moving average using 180 data points (= 3 h) after aligning all time points to clock time and integrated on a 24-h scale. (E) Average RER for all subjects plotted modulo-24 h. Data were averaged into 1-h bins with error bars indicating standard deviation. See S3 Fig for data of all subjects' RER individually plotted. (F) Average hourly pairwise comparison of (breakfast − snack) RER values for all subjects. Error bars indicate 95% confidence intervals, and values are based on a mixed-model analysis. Asterisks indicate significant differences (*p*-value < 0.05) between breakfast and snack values for the indicated 1-h bins. See S2A Table for the hour-by-hour statistical comparison of the Breakfast versus the Snack Sessions and S3 Table for a statistical comparison of peak/trough amplitude. All panels: the blue line indicates values during the subjects' Breakfast Sessions and the red line for the subjects' Snack Sessions. Shading indicates meals and lights off as in Fig 1B and 1C. Error bars indicate ± standard deviation. MR, metabolic rate; RER, respiratory exchange ratio.

The differences in the RER patterns between the two sessions manifest primarily during the time of late-evening snacking and for at least several hours into the sleep episode (hours 22–03 (Fig 2E and S2A Table). Apparently the late-evening snacking delays the temporal switching between primarily carbohydrate-catabolic mode (higher RER values) and primarily lipid-catabolic mode (lower RER values). Despite this change in the daily pattern of RER, the values integrated over the entire 56-h time courses indicate differences slightly above the *p* = 0.05 level between the two sessions in terms of overall RER or total energy expenditure (Fig 2C and 2F; *p* = 0.068 for RER and *p* = 0.11 for MR). Moreover, while there was a significant TEF for MR and RER at each meal, our calculations based on the method of McHill and colleagues [11] indicated no differences in the TEF between the sessions for lunch and dinner—the two meals that were the same in both sessions (*p* = 0.432 for lunch and *p* = 0.855 for dinner). Moreover, the TEF for the breakfast as compared with the snack was also not different (*p* = 0.284). Therefore, differences in MRs as assessed by TEF were not responsible for the substrate-switching preferences that are described below for breakfast skipping versus late-evening snacking.

The conclusion that altered meal timing delays the sleep-onset switching between carbohydrate- and lipid-catabolic modes can be more easily visualized by converting the RER values into CO versus LO rates [26,27]. CO during the Breakfast Session was high during the active day-phase with peaks just after each mealtime, but it dropped precipitously as the subjects entered their sleep episode after lights-out at 11:00 PM (Fig 3A). The CO rate of subjects in the Breakfast Session who had not eaten since dinnertime began to fall before sleep onset and continued to be low through the first half of the nocturnal sleep episode (S2E Table). On the other hand, in the Snack Session, the late-evening snack meal caused a peak carbohydrate catabolism just before going to bed, and while CO dropped thereafter, it remained higher throughout the sleep episode than when the same subjects were in the Breakfast Session (Fig 3A and 3B). Overall, 24-h CO did not differ between sessions because the increased oxidation after breakfast in the Breakfast Session was offset by less CO in the early night of the Breakfast Session (Fig 3C and S2E Table). Therefore, CO was not significantly different between the sessions over the entire 56-h time course ($p = 0.130$).

On the other hand, LO was different between the sessions ($p = 0.028$). Subjects in the Breakfast Session experienced a relatively constant rate of LO throughout the 24-h cycle (Fig 3D and 3E). Because the overall MR declined during the night (Fig 2A), this means that carbohydrate catabolism was "switched off," and LO was sustained during the nightly fast. However, in the Snack Session, the availability of carbohydrates that was enabled by the late-evening snack supported metabolism during the night by CO; since the nocturnal MR is lower than the diurnal MR and carbohydrate catabolism is maintaining nocturnal metabolism, the meal timing of the Snack Session inhibited LO at night (Fig 3C and 3F). On average, 15 more grams of lipid were burned over the 24-h cycle by subjects on the Breakfast Session as compared with the Snack Session (Fig 3F; $p = 0.028$).

Using a mixed-model pairwise hour-by-hour analysis, we found significant differences in both CO and LO in hours 22:00–02:00 of the snack/night interval (S2E and S2F Table), with carbohydrates being utilized at higher rates for a longer time over this temporal window in the Snack Session than in the Breakfast Session (Fig 3B and 3C). Conversely, the Breakfast Session showed significantly more lipids burned in the snack/night interval than did the Snack Session (Fig 3E and 3F). During the breakfast interval, we also found a significant difference in CO, with more carbohydrates burned during the Breakfast Session than during the Snack Session at hours 08:00–09:00 (S2E Table). Nevertheless, over the entire 24-h span, there is not a net difference in CO between sessions because the oxidation difference in the breakfast window is offset by opposite oxidation rates in the snack/night window (Fig 3C). However, the enhanced LO in the snack/night window of subjects in the Breakfast Session is not offset by an opposite effect in another temporal window (Fig 3F). These results indicate that the time of meal placement can cause variation in the amount of lipids oxidized regardless of the nutritional or caloric content of the meal; changing the daily timing of a nutritionally equivalent meal of 700 kcal has a significant impact upon carbohydrate and lipid metabolism.

## Discussion

The major finding of this study is that the timing of feeding over the day leads to significant differences in the metabolism of an equivalent 24-h nutritional intake. Daily timing of nutrient availability coupled with daily/circadian control of metabolism drives a switch in substrate preference such that the late-evening Snack Session resulted in significantly lower LO compared to the Breakfast Session. When the subjects started bedrest after having just eaten the late-evening snack meal (Snack Session), they catabolized less lipid during their sleep episode than they did when they fasted from dinner to breakfast (Breakfast Session). This significant

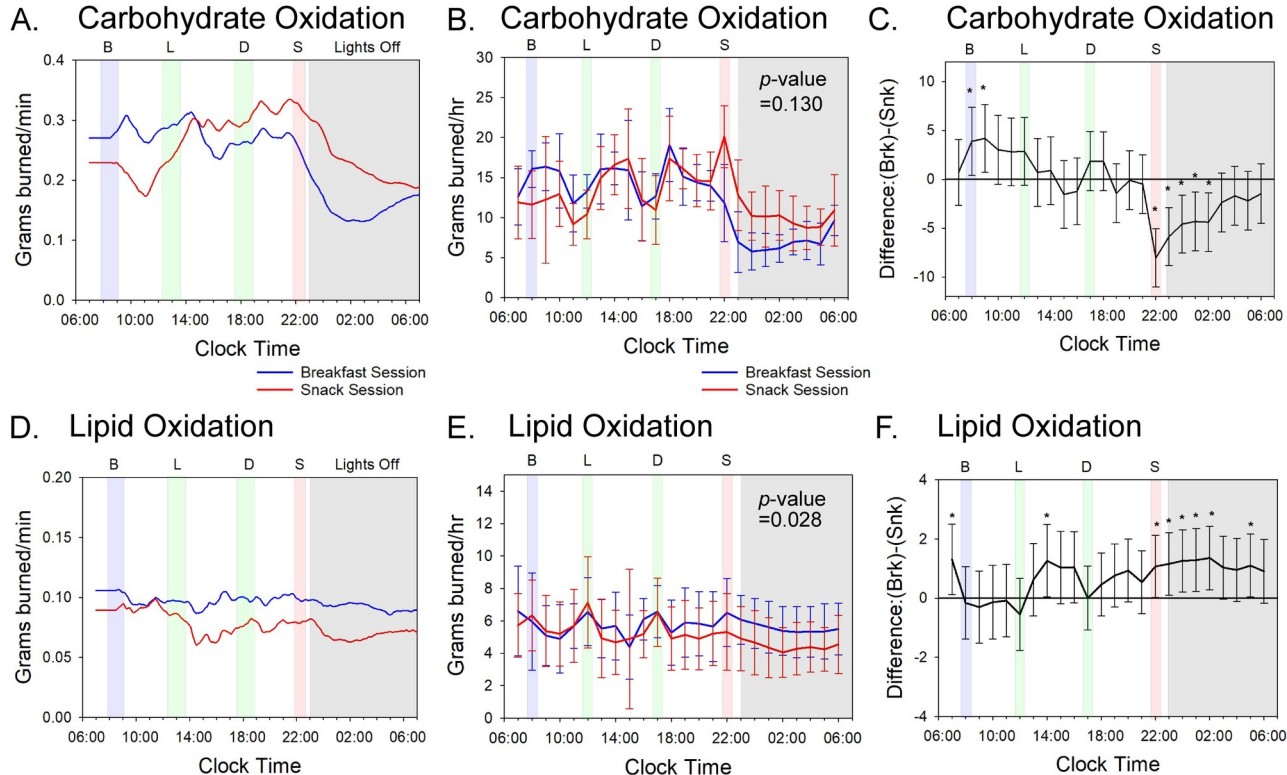

**Fig 3. Meal timing alters substrate oxidation; see S5 and S6 Figs and S2E and S2F Table for underlying data and analyses.** (A) CO data of a representative subject (#3) calculated from indirect calorimetry measurements as described [26,27]. Data are plotted as a 3-h moving average (180-min data points). (B) Average of all subjects for daily CO calculated from indirect calorimetry measurements as described [26,27], and the 56-h time course data are plotted on a modulo-24 h scale. Averaged data for all subjects are organized in 1-h bins. The *p*-value of 0.130 refers to a pairwise comparison of the average (breakfast − snack) difference values over the full 56-h time course for CO. See S5 Fig for data of all subjects' CO rates plotted individually. (C) Average hourly pairwise comparison of (breakfast − snack) difference CO values for all subjects. Error bars indicate 95% confidence intervals, and values are based on a mixed-model analysis. Asterisks indicate significant differences (*p*-value < 0.05) between breakfast and snack values for the indicated 1-h bins. See S2E Table for the hour-by-hour statistical comparison of the breakfast versus the snack sessions. (D) LO data of a representative subject (#3) calculated from indirect calorimetry measurements as described [26,27]. Data are plotted as a 3-h moving average (180-min data points). (E) Average of all subjects for daily LO calculated from indirect calorimetry measurements as described [26,27] and the 56-h time course data are plotted on a modulo-24 h scale. Averaged data for all subjects are organized in 1-h bins. The *p*-value of 0.028 refers to a pairwise comparison of the average (breakfast − snack) difference values over the full 56-h time course for LO. See S6 Fig for data of all subjects' LO rates plotted individually. (F) Average hourly pairwise comparison of (breakfast − snack) difference LO values for all subjects. Error bars indicate 95% confidence intervals, and values are based on a mixed-model analysis. Asterisks indicate significant differences (*p*-value < 0.05) between breakfast and snack values for the indicated 1-h bins. See S2F Table for the hour-by-hour statistical comparison of the breakfast versus the snack sessions. All panels: the blue line indicates values during the subjects' Breakfast Sessions and the red line for the subjects' Snack Sessions. Shading indicates meals and lights off as in Fig 1B and 1C. Error bars indicate ± standard deviation. CO, carbohydrate oxidation; LO, lipid oxidation.

(*p* = 0.028, Fig 3E) effect was measurable over only 3 sleep episodes in our experiments so that an average of 15 fewer grams of lipid were burned over the 24-h cycle by subjects on the Snack Session. The impact of a regular late-evening snack persisting over a longer time would progressively lead to substantially lower LO (and therefore, more lipid accumulation) as compared with fasting during this interval of the day. As schematized in Fig 4, the daily patterns of substrate oxidation (Fig 4A and 4B) are roughly following the daily eating patterns (Fig 4C). However, a late-evening snack likely sustains liver glycogen stores (CO; Fig 4A) so that metabolism does not transition as rapidly or as fully into LO during the nocturnal fast (Fig 4B).

Our interpretation of these data is based on the circadian clock orchestrating a switch between primarily CO to primarily LO between the last meal of the day and the onset of circadian-timed sleep [1,3,6,7]. Instead of fasting between dinnertime and breakfast, if a person eats during the late evening, carbohydrates will be preferentially metabolized as sleep initiates,

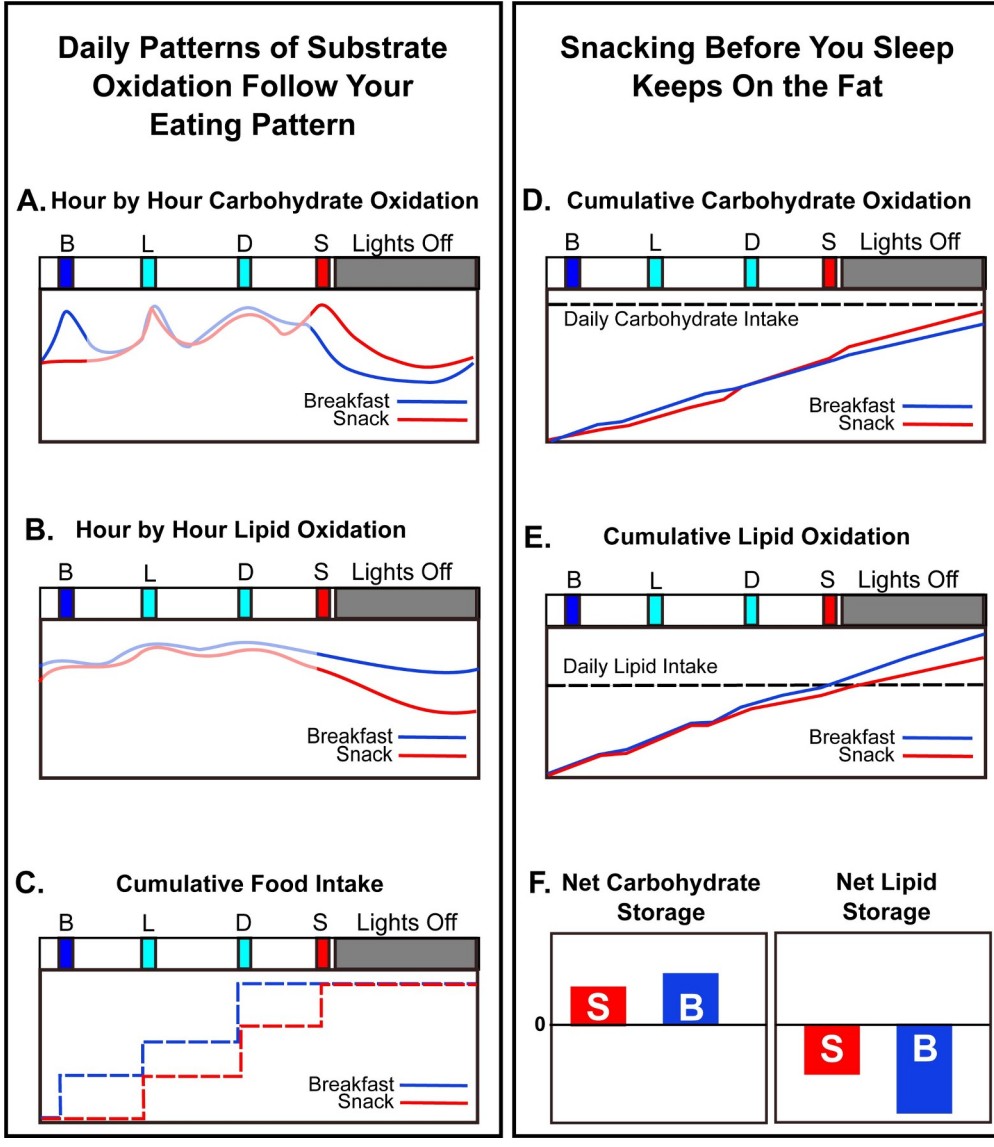

**Fig 4. Schematic: Late-evening snacking interacts with the circadian rhythm of metabolism to inhibit LO.** (A and B) Hour-by-hour oxidation rates for carbohydrates (panel A) and lipids (panel B) in the two sessions. These curves are smoothed versions of the experimental data in Fig 3. (C) Cumulative food intake on the Breakfast versus Snack Sessions. (D and E) Cumulative oxidation rates over the 24-h cycle derived from the curves in panels A and B and the experimental data of Fig 3. Panel D shows cumulative CO, while panel E shows cumulative LO. The horizontal dashed lines indicate the daily total intake of carbohydrates (D) and lipids (E) for comparison with the cumulative respective oxidations. (F) Approximate net relative daily storage of carbohydrates and lipids inferred from the data of Fig 3 and the analyses depicted in the other panels of this figure (arbitrary units). Positive values indicate the extent of substrate accumulation/storage, and negative values indicate the extent of substrate oxidation ("burning"). CO, carbohydrate oxidation; LO, lipid oxidation.

delaying the timing of the switch into primarily LO. Over the 24-h cycle, cumulative CO as compared with total carbohydrate intake was not dramatically different between the two sessions (Fig 4D), so the net 24-h carbohydrate storage is similar (Fig 4F). On the other hand, the cumulative 24-h LO rate as compared with total lipid intake is substantially less when late-evening snacking detains the transition to lipid catabolism (Fig 4E), thereby lessening the mobilization of lipid stores (i.e., the extent of lipids being oxidized; Fig 4F). There is a clear

tradeoff between lipid and CO during the night; the Breakfast Session clearly favors LO at the expense of CO (Figs 3 and 4).

There was a small but significant increase in CO in the morning after eating breakfast (Figs 1D and 3C, hours 08:00–09:00 in S2E Table), but not on LO (Fig 3F, hours 08:00–09:00 in S2F Table). However, the effect of eating versus skipping breakfast is not as significant as the impact of eating after dinner on both carbohydrate and lipid catabolism during the sleep episode (Fig 3). In this study, there were no obvious differences among subjects based on BMI or gender (Table 1 and S3–S6 Figs). Unlike the conclusions of a previous investigation comparing morning versus evening carbohydrate-rich meals [33], in our investigation, the different phasing of the meals between the two sessions did not change the phasing of the daily metabolic pattern. The phasing of sleep during the 56-h time courses matched that of the subjects' sleep patterns for the prior week (compare Table 1 and S2 Fig), and the phasing of the daily rhythms of activity, MR, and CBT were in phase between the two sessions (Fig 2B and S1 Fig). Therefore, in our protocol, the wake/sleep cycle appears to be locked in the same phase relationship to the lights-on/lights-off cycle in both sessions, and the altered meal timing of the Snack Session has delayed the metabolic switching between primarily carbohydrate-catabolism mode and primarily lipid-catabolism modes in relationship to either the circadian system and/or the timing of sleep (Fig 4).

Consistent with the findings of other investigations of altered meal timing, breakfast skipping, time-restricted feeding, etc. [21,22,23], we found no significant differences in total energy expenditure between sessions (Fig 2C). Nevertheless, metabolism was significantly affected. In particular, the average daily RER maintained a higher value in the Snack Session (Fig 2E and 2F), which can be attributed to a delayed entry into primarily LO mode (Figs 2E and 3 and 4). The end result of the reduced LO will be enhanced lipid storage, which over time will lead to increased adiposity. Therefore, in older adults who are potentially at risk for metabolic disorders, avoiding snacking after the evening meal can sustain LO and potentially improve metabolic outcomes.

## Materials and methods

### Ethics statement

The study protocol was approved by the Institutional Review Board of Vanderbilt University's Human Research Protections Program (approval number: 140536) and registered at Clinical-Trials.gov (identifier: NCT04144426). Prior to the study, each subject signed an informed and written consent.

### Subjects

Six subjects (4 male and 2 female) were first recruited through flyers and the Vanderbilt Kennedy Center. Subjects were between 51 and 63 (average age was 57) with BMIs between 22.2 and 33.4 (Table 1). Applicants had to be 50 years of age or older and have no serious health complications or medications that could impact metabolism (see inclusion/exclusion criteria in S1 Text). Female subjects were not required to be postmenopausal for inclusion in this study, but because of the age requirement, all females recruited to the study were postmenopausal. Subjects had no prior shiftwork experience. When interviewed, a questionnaire was administered by the researchers to assess eligibility and sleep/eating habits (see S2 Text). Subjects were recruited October 2015 to February 2017.

Subjects were requested to monitor and record their sleeping habits for the one week prior to each metabolic chamber visit with a log that was provided. Subjects were asked to maintain their regular sleep and feeding schedule for one week prior to the chamber visits.

Serendipitously, all of the subjects maintained a typical sleeping and eating schedule that was approximately in phase with the meal and sleep (lights-off) schedule of the chamber visit (Table 1). Representative examples of the subjects' sleep schedule prior to chamber visits appear in S2 Fig. After the 1-week period, subjects were admitted to the Center for Clinical Research at Vanderbilt University after a health assessment by a physician. Metabolism of the subjects was monitored in the Human Metabolic Chamber at Vanderbilt University, which is a whole-room calorimeter with $CO_2$ and $O_2$ detectors to monitor the rate of $VO_2$ and $VCO_2$ (see S7 Fig). The room had a set flow rate of $O_2$ and $CO_2$ that allowed the energy expenditure of each subject to be measured through indirect calorimetry. Subjects were maintained on an enforced daily light/dark schedule where lights on occurred at 7:00 AM and lights off at 11:00 PM. The subjects ate and slept in the metabolic chamber and were allowed only two brief 20-min episodes per day outside the chamber: once about 10:00 AM to take a quick shower and once about 3:00 PM to take a brief nonstrenuous walk. While in the chamber, the subjects were instructed to do sedentary activities such as reading, internet, watching TV, etc.

During both visits, the Vitalsense Integrated Monitoring Physiological System was used to monitor CBT over the course of the study. Subjects were given a Vitalsense telemetric CBT capsule that recorded the subject's CBT and relayed information to a monitor attached to the subject's waistband (or under the pillow during lights off). Telemetric capsules were given every 24 h to maintain consistent temperature readings independent of bowel movements when the sensor might be excreted. Meals and lights-off times were scheduled regularly as shown in Fig 1A. Subjects were admitted into the metabolic chamber for 2.5 days, starting at 5:30 PM and ending 7:00 AM after the third night in the facility. Subjects were admitted for two separate crossover sessions at the facility with a shifted meal schedule. For one session (the Breakfast Session), subjects were given a breakfast at 8:00 AM, lunch at 12:30 PM, and dinner at 5:45 PM every day. For the other session (the Snack Session), subjects were given decaf coffee (without cream or sugar but including an artificial sweetener if desired) at 8:00 AM, lunch at 12:30 PM, dinner at 5:45 PM, and a late-evening snack meal at 10:00 PM. The breakfast on day 2 was identical to the snack on day 2, and the breakfast on day 3 was identical to the snack on day 3 (S1A Table), and each was approximately 700 kcal. The menus of the meals are shown in S1A Table. The order of the sessions (i.e., Breakfast Session first versus Snack Session first) was determined in a randomized design for each subject (Table 1), and 4–12 days elapsed between sessions, depending upon the subject. Subjects were asked to eat all the meal provided, but any leftover food was weighed back, and actual intake for each meal is shown in S1B and S1C Table. The size of the meals was determined by nutritionist to account for calories burned for each individual (on average, a daily 2,300 kcal diet). Calories were divided as follows: approximately 700 kcals for Breakfast/Snack, approximately 600 kcals for Lunch, and approximately 1,000 kcals for Dinner.

## Whole-room respiratory chamber

The room calorimeter at Vanderbilt University is an airtight room (17.9 m$^3$) providing an environment for daily living whose accuracy has been documented (S7 Fig [34]). The room has an entrance door, an airlock for passing food and other items, and an outside window. The room is equipped with a TV/media system, toilet, sink, desk, chair, and rollaway bed allowing overnight stays. The calorimeter is located in the Clinical Research Center at Vanderbilt University, and an intercom connects the chamber to a nearby station where nurses are on duty 24 h/7 days per week. Temperature, barometric pressure, and humidity of the room are controlled and monitored. Minute-by-minute energy expenditure (kcal/min) are calculated from

measured rates of $O_2$ consumption and $CO_2$ production using Weir's equation [35]. For raw $VO_2/VCO_2$ respiratory data, see S1 Data.

## Quantification and statistical analyses

To quantify the differences between the Breakfast and Snack Sessions, we applied both a paired *t* test and a linear mixed model to the full 56-h time course for each of the following measurements: MR, activity, CO, LO, RER, and CBT. Each measurement was averaged using hourly bins for each subject in each session. By using a mixed model, we were able to adjust for dependency of within-subject observations. The model included random intercept and fixed effects for session (Breakfast versus Snack), day (treated as a factor variable and defined as 3:00 PM on one day to 2:59 PM on the next day), hour (treated as a factor variable), an interaction between session and day, and an interaction between session and hour. If the *p*-value of an interaction was greater than 0.2, we removed that interaction from the model.

For an hourly paired *t* test, a sample size of 6 has 80% power to detect an effect size of 1.435 (defined as mean difference/standard deviation) with a 0.050 two-sided significance level. We applied a mixed model to perform an integrated analysis combining all hourly measurements together. The mixed model is more powerful than a traditional *t* test comparison, such that the effective sample size of our study calculated from the mixed model is approximately 21 [36].

## Calculations

Daily CO and daily LO were calculated from indirect calorimetry measurements as described [26,27]. Nitrogen excretion rate was based on the amount of protein provided to subjects as well as previous research that monitored 24-h nitrogen using similar parameters [37]. Because protein content was not altered between sessions, we assumed that 24-h nitrogen excretion rate was equivalent for Breakfast versus Snack Sessions.

## Supporting information

**S1 Fig. Activity and CBT patterns.** See S2B and S2C Table for underlying data. (A) Average wrist locomotor activity (measured in arbitrary units based on vector of magnitude) of all subjects for the 56-h time course. The blue line indicates values during the subjects' Breakfast Sessions and the red line for the subjects' Snack Sessions. Black arrows indicate the afternoon break during which subjects were allowed to exit the chamber for a 30-min break, during which the subjects were allowed a nonstrenuous walk. (This 30-min interval was excluded in other measurements because calorimetric readings were not being taken during this break.) (B) Average wrist activity of all subjects plotted modulo-24 h. Minute-by-minute activity data were averaged for all subjects into 1-h bins and aligned by clock time. The arrow denotes the 30-min break referred as noted in panel A. The *p*-value of 0.538 refers to a pairwise comparison of the average ([breakfast] − [snack]) difference values over the full 56-h time course for wrist activity. See S2B Table for the hour-by-hour statistical comparison of the breakfast versus the snack sessions. (C) Average CBT for all subjects over the 56-h time course. (D) Average CBT of all subjects plotted modulo-24 h. Minute-by-minute activity data were averaged for all subjects into 1-h bins and aligned by clock time. The *p*-value of 0.218 refers to a pairwise comparison of the average ([breakfast] − [snack]) difference values over the full 56-h time course for CBT. See S2C Table for the hour-by-hour statistical comparison of the breakfast versus the snack sessions. All panels: the blue line indicates values during the subjects' Breakfast Sessions, and the red line the values for the subjects' Snack Sessions. Shading indicates meals and lights off as in Fig 1B and 1C. Error bars indicate ± standard deviation (*n* = 6). CBT, core body temperature. (TIF)

**S2 Fig. Subject's self-reported schedules prior to entry into experiment.** The subjects' self-reported bedtime and wake-up time for the week prior to entry into the metabolic chamber shows that the daily phasing of sleep was similar before and during the 56-h experimental time course. Black squares specify the time of bedtime and wake-up, with the horizontal lines indicating sleep episodes prior to entry into the metabolic chamber (blue horizontal lines) or during the 56-h experimental time course (red horizontal lines). Therefore, the subjects did not experience a phase shift of their daily cycle when they entered the experimental conditions in the metabolic chamber (compare with Table 1).
(TIF)

**S3 Fig. Individual daily RER data.** Average RER ($VCO_2/VO_2$) values for subjects 1–6 from their Breakfast Session (blue) and Snack Session (red) averaged into 1-h intervals. Error bars indicate standard deviation. RER, respiratory exchange ratio.
(TIF)

**S4 Fig. Individual daily MR data.** Hourly kcals burned for subjects 1–6 from their Breakfast Session (blue) and Snack Session (red) averaged into 1-h intervals. Error bars indicate standard deviation. MR, metabolic rate.
(TIF)

**S5 Fig. Individual daily CO.** Hourly grams of carbohydrates burned for subjects 1–6 from their Breakfast Session (blue) and Snack Session (red) averaged into 1-h intervals. Daily CO values calculated from indirect calorimetry measurements as described [26,27]. Error bars indicate standard deviation. CO, carbohydrate oxidation.
(TIF)

**S6 Fig. Individual daily LO.** Hourly grams of lipids burned for subjects 1–6 from their Breakfast Session (blue) and Snack Session (red) averaged into 1-h intervals. Daily LO values calculated from indirect calorimetry measurements as described [26,27]. Error bars indicate standard deviation. LO, lipid oxidation.
(TIF)

**S7 Fig. Configuration and photographs of the human whole-room calorimetry chamber at Vanderbilt University.**
(TIF)

**S1 Table.** (A) Representative meals, (B) nutritional information, (C) nutritional information for each subject.
(XLSX)

**S2 Table.** Hour-by-hour mixed-model analyses for (A) RER, (B) activity, (C) CBT, (D) MR, (E) CO, and (F) LO. CBT, core body temperature; CO, carbohydrate oxidation; LO, lipid oxidation; MR, metabolic rate; RER, respiratory exchange ratio.
(XLSX)

**S3 Table. Mixed-model analysis of peak/trough amplitude.** The mixed-model analysis was applied to the difference between the fitted maximum and the fitted minimum across the 24-h day for each physiological parameter. The table lists the *p*-value from the mixed model to determine that there were no statistically significant changes in peak/trough amplitude between Breakfast and Snack Sessions in any of the measured parameters.
(XLSX)

**S1 Data. Raw VO₂ and VCO₂ data from the whole-room respiratory chamber.**
(XLSX)

**S1 Text. Inclusion/exclusion criteria and CONSORT flow diagram for randomized trials.**
(DOCX)

**S2 Text. Questionnaire for subject recruitment.**
(DOCX)

**S1 Checklist. CONSORT checklist.**
(DOC)

## Acknowledgments

We dedicate this study to the memory of Dr. Martin Katahn, a pioneer of the Rotation Diet and the benefactor whose donation enabled the metabolic chamber at Vanderbilt University. The authors would like to thank Briana Wyzinski, Ian Dew, and Regina Tyree for their contributions to subject recruitment and respiratory chamber operation. We thank Cynthia Dossett, Holly Mason, and Dr. Heidi Silver for their help with nutrition information and meal development. We also thank the subjects who participated in this study.

## Author Contributions

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
