## [Editor Report · Decision Letter 0]

18 Nov 2019

Dear Dr Johnson, 

Thank you for submitting your manuscript entitled "Eating breakfast and avoiding the evening snack sustains lipid oxidation" for consideration as a Research Article by PLOS Biology.

Your manuscript has now been evaluated by the PLOS Biology editorial staff as well as by an Academic Editor with relevant expertise and I am writing to let you know that we would like to send your submission out for external peer review. Please note, however, that the outcome of our discussion of your manuscript is that we have some reservations as to the strength of analysis given the low sample size. We would need to be persuaded by the reviewers that the paper has the potential to offer the significant strength of advance and strong support for the conclusions in order to pursue it further for PLOS Biology after review, and will likely require a larger sample size. In addition, given the nature of the manuscript, we would like to review your paper as a Short Report, and not as a Research Article. Therefore, I ask that you please select the Short Report article type when completing the full submission and providing the information outlined below. 

Before we can send your manuscript to reviewers, we need you to complete your submission by providing the metadata that is required for full assessment. To this end, please login to Editorial Manager where you will find the paper in the 'Submissions Needing Revisions' folder on your homepage. Please click 'Revise Submission' from the Action Links and complete all additional questions in the submission questionnaire.

Please re-submit your manuscript within two working days, i.e. by Nov 20 2019 11:59PM.

Kind regards,

Hashi Wijayatilake, PhD,

Managing Editor

PLOS Biology

---

## [Decision Letter · Decision Letter 1]

20 Dec 2019

Dear Carl,

Thank you very much for submitting your manuscript "Eating breakfast and avoiding the evening snack sustains lipid oxidation" for consideration as a Short Reports by PLOS Biology. As with all papers reviewed by the journal, yours was evaluated by the PLOS Biology editors as well as by an Academic Editor with relevant expertise and by independent reviewers. As you can see from the reviews appended below, the reviewers are all very positive about the manuscript. They have relatively minor requests for revision including changing terminology, adding references, adding more methodological details, and better discussing and caveating confounds and alternate explanations. Reviewer 2 does also ask for some additional analysis of the data. Regarding sample size and statistics - I have discussed your explanation regarding study design and the larger effective sample size in detail with the journal's statistics expert. The expert agreed with the methodological arguments and noted that the design and corresponding analyses do what they promise. Given the reviewer comments on the crossover design and the statistic expert's detailed comments to us, the Academic Editor and I are reassured on this point. Please do however address Reviewer 3's points about clarifying the statistics and power calculation. Overall, based on the reviews, we will probably accept this manuscript for publication, assuming that you will modify the manuscript to address the points raised by the reviewers. 

Important: Please also make sure to address the data and other policy-related requests noted at the end of this email.

We expect to receive your revised manuscript within two weeks. Your revisions should address the specific points made by each reviewer. In addition to the remaining revisions and before we will be able to formally accept your manuscript and consider it "in press", we also need to ensure that your article conforms to our guidelines. A member of our team will be in touch shortly with a set of requests. As we can't proceed until these requirements are met, your swift response will help prevent delays to publication.

*Copyediting*

*Published Peer Review History*

*Early Version*

*Submitting Your Revision*

Sincerely,

Hashi Wijayatilake, PhD, 

Acting Chief Editor

PLOS Biology

ETHICS STATEMENT:

The Ethics Statements in the submission form and Methods section of your manuscript should match verbatim. Please ensure that any changes are made to both versions.

-- Please include the full name of the IACUC/ethics committee that reviewed and approved the animal care and use protocol/permit/project license. Please also include an approval number.

-- Please include the specific national or international regulations/guidelines to which your animal care and use protocol adhered. Please note that institutional or accreditation organization guidelines (such as AAALAC) do not meet this requirement.

-- Please include information about the form of consent (written/oral) given for research involving human participants. All research involving human participants must have been approved by the authors' Institutional Review Board (IRB) or an equivalent committee, and all clinical investigation must have been conducted according to the principles expressed in the Declaration of Helsinki.

SPECIAL NOTES:

> Please read through this page in detail for our policy and documentation requirements related to human subject research:

https://journals.plos.org/plosbiology/s/human-subjects-research

> Please report details on how informed written consent for the research was obtained and confirm that the individuals have provided written consent for the use of that information. Please add this to the Ethics Statement.

> Please also provide CONSORT checklists (or TREND) and other documentation as supplemental files, as relevant.

DATA POLICY:

**Thank you for providing the excel files with the Dexa results, nutritional data and VO2/VCO2 values. Please confirm that these datasets contain all the individual numerical values that underlie the summary data displayed in the following figure panels as they are essential for readers to assess your analysis and to reproduce it:

Figs. 1BCD, 2A-D, 3A-F, 4A-C, 5A-C and similarly all supplemental figures.

REVIEWS:

Reviewer #1: 

Congratulations on a very interesting study! Overall, I think this paper by Kelly et al., provides a novel understanding of how the phase of eating alters metabolism, specifically lipid oxidation. The use of the metabolic chamber for 56 hours provides very robust data and as far as I am aware, has not yet been used to understand changes in eating phase in older, overweight adults. As the timing of eating is becoming an increasingly exciting field of study, many researchers and medical professionals will be interested in these findings as it provides novel insights into the importance of the phase of eating. I recommend that this paper is published with the minor revisions suggested below. 

1. My largest concern is about the terminology used to describe the meals. First, breakfast is a very vague term and does not have a strong definition. For instance, NHANES surveys define breakfast as anything consumed before breakfast, during breakfast, or before lunch. Because of the subjective nature of the term breakfast, some readers (especially the media and general public) may take breakfast to mean very different things, including very early morning meals right as they wake up, which is not what was tested. 

Secondly, the term ‘snack’ implies that it is a smaller meal. The authors do make it clear in the manuscript that they are the same number of calories, but for many that will not read the whole paper, this will likely be misleading, and should instead be referred to as a late-night meal. 

Due to these likely misconceptions, I feel that changing the title and terminology throughout the text to refer to the different eating patterns as daytime and late-night eating, or early and late/delayed eating (as other papers have done – Hutchison et al., 2019) will more clearly explain the research. Additionally, the title is misleading as this paper does not fully determine if it is the presence of the breakfast, or the lack of night meal that leads to better lipid oxidation, but rather the phase of eating that made the difference. I agree the way that it is written is technically correct, but I believe it will be misinterpreted as it can be avoided by changing the wording.

2. There are multiple papers that are relevant to this work that have not been cited and I would encourage adding to the introduction and/or discussion as the authors see fit. 

• Early vs Delayed TRE: 

o Hutchison, A.T., Regmi, P., Manoogian, E.N., Fleischer, J.G., Wittert, G.A., Panda, S. and Heilbronn, L.K., 2019. Time‐Restricted Feeding Improves Glucose Tolerance in Men at Risk for Type 2 Diabetes: A Randomized Crossover Trial. Obesity, 27(5), pp.724-732.

• Timing of meals on metabolic health

o Jakubowicz, D., Wainstein, J., Landau, Z., Ahren, B., Barnea, M., Bar-Dayan, Y. and Froy, O., 2017. High-energy breakfast based on whey protein reduces body weight, postprandial glycemia and hba1c in type 2 diabetes. The Journal of nutritional biochemistry, 49, pp.1-7.

o Many other papers by Oren Froy’s group

• Breakfast skipping and late-night eating are both associated with increased risk of coronary heart disease (good example of how late-night eating had a larger effect, but breakfast skipping was overemphasized as the cause of increased risk – need to make sure this does not happen here).

o Cahill, L.E., Chiuve, S.E., Mekary, R.A., Jensen, M.K., Flint, A.J., Hu, F.B. and Rimm, E.B., 2013. Prospective study of breakfast eating and incident coronary heart disease in a cohort of male US health professionals. Circulation, 128(4), pp.337-343.

3. Please clarify in the methods if artificial sweeteners were allowed in the black decaf coffee.

4. In the text and figure 1 it shows that the snack was given at 10pm and lights off at 11pm, but in Supplementary Table 1 says snack is given at 11pm. Please correct or clarify.

5. Unlike the sleep data in the week leading up the metabolic chamber, the data on food intake leading up the metabolic chamber sessions is not provided in the supplement and is never mentioned except for the fact that they had them record it. Please provide a few sentences commenting on the eating pattern of the participants leading up to the trail. What was their eating window/when did they eat? How did the calories compare to the calories consumed while in the metabolic chamber? Etc. 

6. Add units Fig 2A and B Activity – is it just A.U.? Also, for the comparison, was it total activity counts that was compared?

7. States that sleep quality equivalent in both groups, but there are no stats of sleep efficiency provided. Please add and/or clarify how this was determined.

8. Be consistent about how figures are referenced in the text. Sometime figures are referred to by number and letter, sometimes only by number even when specifically referring to letters.

9. Some statements in the results seems more like discussion and makes assumptions about mechanism that were not directly tested, such as: “Apparently the late-evening snacking delays the clock induced switching between primarily carbohydrate-catabolic mode (higher RER values) and primarily lipid-catabolic mode (lower RER values.” Because clock control was not directly tested, this should not be stated as a result. Please reword.

----

Reviewer #2: 

This is an excellent short report from Parsons Kelly and colleagues. Although preliminary, it fits well with the ethos of the short report article type. The data are presented well and completely, along with helpful supplementary data to allow further assessment of the raw data if necessary. 

The principal conclusion of the report is that having an evening snack inhibits lipid oxidation (compared to having those calories at breakfast time), and in the long term, in metabolically "at risk" individuals, this could result in cumulative adipose tissue mass over time and the attendant effects on whole-body metabolism and notably obesity and diabetes.

The authors comment that RER is different between the two groups in the sleep phase, but not at other times. Other parameters (e.g. activity) are not different. However, i think that it would be interesting to analyze the data in a bit more depth than using a mixed model to detect differences between the groups. There doesn't appear to be any analysis of diurnal variation in the two groups. For example, comparing Fig. 1B and 1C, they look quite different over time, with the snack group (Fig. 1C) demonstrating a lower amplitude diurnal rhythm that the breakfast group (Fig. 1B). It would be instructive to estimate amplitude of these cycles (e.g. by using Circwave, JTK_Cycle, RAIN, or even PLS fitting). The authors should incorporate any arising points in their discussion. Other than this point, the analysis is very good, and the cross-over design is welcomed.

--

Reviewer #3: 

This is an excellent and timely piece of research and clearly merits publication in a broad interest journal such as PLoS Biology.

Over the last ten years there has been enormous interest in the idea that, in addition to energy expenditure and what you eat, when you eat can have important effects on how food is metabolized and stored. There are a number of compelling mouse studies in this area, with correlative evidence from humans, but a clear interventional test of this phenomenon in humans under a well-controlled but pseudo real-world setting has been lacking. The authors have performed this study, using metabolic chambers and an elegant crossover design, in healthy middle-aged individuals; this being particularly pertinent as it reflects the demographic group most at risk from obesity in their country.

The data presented boils down to the analysis of a single experiment where oxygen consumption, carbon dioxide production, physical activity and core body temperature was measured with high temporal resolution over two sets of 3 days. Food intake was isocaloric between sessions, and the only difference was breakfast vs. bedtime snack, with the key finding being that bedtime snacks attenuated the nighttime dip in respiratory exchange ratio compared with the breakfast session, indicative of reduced lipid oxidation at night when feeding occurs later in the diurnal cycle. This data are convincing, with important implications for making achievable lifestyle interventions in obese individuals, and so important finding needs to be communicated to a wide audience. 

Because this paper is likely to find a wide readership, there are some points of interpretation and presentation that I believe warrant some further consideration, as follows:

1) "The phasing or amplitude of the daily rhythms of master clock markers (plasma melatonin and cortisol), insulin, or plasma triglycerides is also not responsible, as reported by Wehrens and coworkers who found no differences in those rhythms in a meal timing study using a very similar protocol to ours [30]."

…I think this sentence is problematic for three reasons.

Firstly, the design of study by Wehrens is very different to this study i.e. their data were collected from healthy young individuals under constant routine with frequent isocaloric snacks i.e. no feeding, sleep or environmental rhythms, which occurred after and not during the scheduled feeding protocol and is therefore highly dissimilar from this protocol. Indeed the absence of a feeding rhythm in Wehrens et al is evident from the Fig2B of that paper, where insulin levels are continuously elevated and show far less temporal variation than normally occurs under physiological, diurnal conditions.

Secondly, Wehrens do in fact report a different acrophase of insulin (p=0.029), but appear to determine that this is not significant for reasons which are opaque to me.

Thirdly, signalling through the insulin and related receptors is the primary established mechanism that determines whether tissues store or mobilise stored carbohydrates and lipids. Serum insulin is more sensitive to dietary carbohydrates, free insulin-like growth factor concentration is more sensitive to dietary protein and fat, with both increasing after a balanced meal and much overlap at the peptide and receptor level (Livingstone et al, 2013; Slaaby et al., 2006). It would therefore be very much at odds with my understanding of physiology if there was no difference in the 24h profile of plasma insulin and other feeding-related endocrine cues between the breakfast and bedtime snack sessions. If the authors wish to discount changes in insulin signalling from contributing the differences in RER observed between the two regimes, then they need to show some data to support this or they need to cite a more compelling and appropriate set of references. Alternatively, they may wish to simply wish to remove mention of insulin at this point in the manuscript and instead speculate about its potential relevance in the discussion. If I have misunderstood their argument, in which case please could they frame it more clearly.

2) Statistics - the description of the linear mixed model, its parameters and implementation needs to be described in more detail. I assume that this more sophisticated analysis provides greater power, but it would be reassuring to the less-statistically literate to know whether a simple repeated-measures two-way ANOVA (regime vs time interaction) agrees with the linear mixed model. Moreover, it is good practice to report the power calculation that was performed before the experiment, which suggested that 6 individuals would be sufficient to detect a difference. 

3) Analysis - by reporting the mean RER (e.g. Fig1B), the authors do not take into account any intra-individual variation in average RER. This is not a problem, but it seems plausible to me that one person might not have exactly the same basal RER are as somebody else due to genetic/epigenetic factors. Thus, if any variation exists, it does not seem unreasonable to me to correct for small differences in average RER (over the six days) between individuals as this would potentially remove a source of systematic error which is not relevant to their hypothesis, and thereby improve their sensitivity to detect differences that are specific to the two different feeding regimes. 

4) Interpretation - dietary carbohydrates and lipids are used for many purposes besides respiration which are not explicitly considered. My understanding is that the major destinations of dietary carbohydrate are glycolysis+respiration (consumes O2, produces CO2) , glycolysis only (no O2/CO2 change), biosynthesis by pentose phosphate pathway (generates CO2, no O2 consumption), protein/lipid glycosylation (no O2/CO2 change), glycogen synthesis (no O2/CO2 change); whereas most dietary lipids are used for respiration via beta-oxidation (consumes O2, produces CO2), stored in lipid droplets (no O2/CO2 change), or used for membrane biosynthesis (no O2/CO2 change). RER would also be affected by any diurnal or feeding related changes in blood pH I imagine.

Thus, whilst it is entirely plausible that the changes in RER ratio over 24h, and between feeding regimes, mean exactly what the authors suggest they mean, there are several potential confounds to this interpretation which cannot be explicitly discounted without further experiments. Whilst this could be achieved by metabolic flux labelling with 13C-glucose, in no way would I expect the authors to perform any additional experiments since this would not affect the primary observation, that the night-time dip in RER is attenuated by a bedtime snack, only its interpretation. I would therefore suggest that some additional caveats to their preferred interpretation would be appropriate in the discussion section. 

Indeed, their model (Fig6) does not really distinguish between the relative proportions of dietary vs stored carbohydrates/lipids that are being utilized as a function of time vs. feeding time - I would personally prefer that they propose a more detailed model that makes some explicit and testable predictions, which could be followed up in further rodent/human experiments, but the authors are free to disagree with my personal taste of course.

---

## [Editor Report · Decision Letter 2]

24 Jan 2020

Dear Dr. Johnson,

On behalf of my colleagues and the Academic Editor, Dr. Achim Kramer, I am pleased to inform you that we will be delighted to publish your Short Reports in PLOS Biology. 

Early Version

PRESS 

Kind regards,

Krystal Farmer,

Development Editor

PLOS Biology

on behalf of

Hashi Wijayatilake,

Managing Editor

PLOS Biology